# Thermoplastic Starch with Poly(butylene adipate-*co*-terephthalate) Blends Foamed by Supercritical Carbon Dioxide

**DOI:** 10.3390/polym14101952

**Published:** 2022-05-11

**Authors:** Chih-Jen Chang, Manikandan Venkatesan, Chia-Jung Cho, Ping-Yu Chung, Jayashree Chandrasekar, Chen-Hung Lee, Hsin-Ta Wang, Chang-Ming Wong, Chi-Ching Kuo

**Affiliations:** 1Institute of Organic and Polymeric Materials, Research and Development Center of Smart Textile Technology, National Taipei University of Technology, Taipei 10608, Taiwan; allan6730@gmail.com (C.-J.C.); manikandanchemist1093@gmail.com (M.V.); jayashreechem7@gmail.com (J.C.); htwang@mail.ntut.edu.tw (H.-T.W.); 2Institute of Biotechnology and Chemical Engineering, I-Shou University, Kaohsiung 84001, Taiwan; isu10605060a@cloud.isu.edu.tw; 3Division of Cardiology, Department of Internal Medicine, Chang Gung Memorial Hospital-Linkou, Chang Gung University College of Medicine, Taoyuan 33305, Taiwan; 4CoreTech System Co., Ltd., Hsinchu 30265, Taiwan; cmwong@moldex3d.com

**Keywords:** thermoplastic starch, silane, foam, carbon dioxide

## Abstract

Starch-based biodegradable foams with a high starch content are developed using industrial starch as the base material and supercritical CO_2_ as blowing or foaming agents. The superior cushioning properties of these foams can lead to competitiveness in the market. Despite this, a weak melting strength property of starch is not sufficient to hold the foaming agents within it. Due to the rapid diffusion of foaming gas into the environment, it is difficult for starch to maintain pore structure in starch foams. Therefore, producing starch foam by using supercritical CO_2_ foaming gas faces severe challenges. To overcome this, we have synthesized thermoplastic starch (TPS) by dispersing starch into water or glycerin. Consecutively, the TPS surface was modified by compatibilizer silane A (SA) to improve the dispersion with poly(butylene adipate-co-terephthalate) (PBAT) to become (TPS with SA)/PBAT composite foam. Furthermore, the foam-forming process was optimized by varying the ratios of TPS and PBAT under different forming temperatures of 85 °C to 105 °C, and two different pressures, 17 Mpa and 23 Mpa were studied in detail. The obtained results indicate that the SA surface modification on TPS can influence the great compatibility with PBAT blended foams (foam density: 0.16 g/cm^3^); whereas unmodified TPS and PBAT (foam density: 0.349 g/cm^3^) exhibit high foam density, rigid foam structure, and poor tensile properties. In addition, we have found that the 80% TPS/20% PBAT foam can be achieved with good flexible properties. Because of this flexibility, lightweight and environment-friendly nature, we have the opportunity to resolve the strong demands from the packing market.

## 1. Introduction

In the past few decades, many industries were developed to manufacture conventional plastics for daily products and polymer-based energy and AI technologies. Particularly, these included multifunctional movement and pressure sensors [1,2], optoelectronic devices [3,4,5], and wearable electronic devices [6,7,8]. However, these plastic products required several hundred years to fully decay into the soil [9]. Furthermore, their collection and storage for recycling is an undesirable and time-consuming process [10]. A viable strategy is to develop bio-based polymer materials as alternatives to these materials to minimize the use of non-biodegradable polymers [11,12,13,14,15]. Considering these, the key to biodegradable polymer-based foams can be used in a wide range of applications, owing to their biocompatibility [16,17], biodegradability, and renewability process [18].

Starch is an eco-friendly biopolymer, with low costs and is easily obtained, although the processing is complicated and difficult to purify. Thermoplastic starch (TPS) can be mixed with water or plasticizers to improve its processability. TPS has an opportunity to be processed by extrusion, injection, or molding to make films, sheets, foams, etc.

There are many works associated with TPS, such as the study of starch-based films [19,20,21,22,23,24,25,26,27,28], the production of innovative biodegradable products [29], the additives used to strengthen TPS [30], water or azodicarbonamide used as foaming agents to develop starch-based foams [18,26,31,32,33,34,35,36,37,38], and evaluating the performance of starch loose-fill foams and expanded polystyrene [39]. Foam products are often used in packing goods, which can easily become plastic garbage. Loose-fill foams made with biodegradable materials, such as TPS or TPS mixed with other plastics, have received attention recently.

Starches with high amylose content are able to produce TPS foams [18,26,31,32,35,36,38]. Starch grafting by poly(methyl acrylate) [34], or grafting polystyrene [37] using water as a foaming agent to make foams, has been studied. Due to the poor mechanical properties of TPS, it is not suitable for producing foam materials, whereas TPS can be blended with other materials to improve properties in order to make the composite foams. However, TPS is a hydrophilic material, which is not suitable for hydrophobic polymeric resins. Therefore, surface modification is necessary to enhance the compatibility between two materials. In general, fatty acids are preferred for generating the peroxide or hydroperoxide reactive species to attack the carbon linkages in the starch to blended with polystyrene (PS), polyolefin, thermoplastic polyurethane (TPU), and polylactide (PLA) with free-radical grafting of maleic anhydride mixed with TPS, etc., [40,41]. Maleated TPS with PBAT [42,43,44,45] or TPS with maleated PBAT [46] are the materials for research.

PBAT is a soft and biodegradable co-polyester synthesized from fossil resources, which is eco-friendly, easily processable, and has a high elongation at break in tensile tests. PBAT can also be used for goods packaging applications, even though the price is not affordable for daily usage. PBAT blended with TPS can lower the material price to possibly manufacture acceptable products and vice versa, TPS mixed with PBAT can improve TPS mechanical properties. In the aforementioned works regarding foaming, with the emergence of various social and environmental concepts, water is the major foaming agent for the TPS foaming process. Although it is difficult for TPS to be foamed under supercritical CO_2_ as a foaming agent, we have addressed these disadvantages by optimizing the ratio between PBAT and foaming agents by using surface modification techniques.

In this study, we used an inexpensive industrial starch with water and glycerol added to become thermoplastic starch (thermoplastic starch, TPS), which was the main foaming material. Further, combined with innovative chemical modification technology, the melt strength was improved to solve the problem of the insufficient structural melt strength of thermoplastic starch. By expanding the molecular chains of TPS/biodegradable polyester composites and generating intermolecular entanglement with each other, the composites have a certain elasticity and buffering properties, which can support the cell structure during the cell growth process. It also has the advantages of energy saving and carbon emission reduction because of its bio-plastic foam material, which can be decomposed and recycled, thus solving the problem of white pollution caused by petrochemical plastics. This starch-based material and processing technique can bring about a new generation of materials that will inevitably develop eco-friendly applications.

## 2. Experimental Preparations

### 2.1. Materials and Procedure

There are three materials, i.e., tapioca starch/TPS (thermoplastic starch: from Roi Et Group, Yannawa, Thailand), PBAT (Poly(butylene adipate-co-terephthalate) (Ecoflex: from BASF, Lemförde, Germany), and Silane A 6040 (SA: from Ya-Hu-Chi industrial Co., Zhubei, Taiwan) that were used in the study. SA has three methoxy groups and one epoxy group; the molecular weight of SA is 0.236 kg/mole and the boiling point is 190 °C. Starch was first mixed with water and glycerin to form TPS and then the TPS surface was modified using SA for the feasible blending with PBAT. The blending ratios with symbols are displayed in Table 1**.** The sheet of PBAT, TPS/PBAT, and (SA/TPS)/PBAT blends were produced by a hot press at a temperature of 140 °C. The sheet area was 75 mm × 75 mm × 3 mm (thickness) and these sheets were used for foaming by CO_2_ under supercritical conditions. In the above, the mixing, preparation, and foaming of this experiment were repeated five times to ensure a high degree of accuracy in the results, and it has excellent reproducibility.

### 2.2. Functional Group Chemical Modification TPS Experiment

The functional group chemical modification of the TPS experiment: First, the commercially available starch raw materials, water, and appropriate glycerin were kneaded in a mixing machine plastic spectrometer (Barbender MIX, Kulturstraße, Germany) to make the starch uniformly thermally plasticized into TPS. Then, a functional group modifier-coupling agent (SA) for TPS modification was added. The reaction temperature was 70~55 °C, and the reaction time was 30~60 min. The pelletizer was modified and pelletized to produce surface-modified SA/TPS blends.

### 2.3. The (SA/TPS)/PBAT Biodegradable Polyester Composite Mixing

The modified (SA/TPS) weight ratio of 50, 60, 70, and 80% was added with different proportions of 50, 40, 30, and 20% biodegradable polyester (PBAT) and then kneaded by the mixing machine plastic spectrometer. The SA/TPS and the biodegradable polyester are uniformly mixed and dispersed into the composite. The mixing machine plastic spectrometer temperature is 90 °C to 145 °C, and the granulation screw speed is 50~100 rpm. Similarly, a control sample of unmodified TPS was blended with PBAT.

### 2.4. The (SA/TPS)/PBAT Composite Foam Test Piece and Supercritical Foaming Experiment

Primarily, the (SA/TPS)/PBAT composite particles entered the hot press (DAKE, Grand Haven, MI, USA), and were hot-pressed at a temperature of 140 °C to 165 °C, then, a square test piece of 75 mm × 75 mm × 3 mm size was retrieved.

As the prepared composite square test piece was placed into the mold of supercritical foaming equipment (200-ton capacity, Tainan City, Jing Day Machinery Industrial Co., Ltd., Taiwan) and obtained incompatible foams according to the different foaming parameter settings, the manufactured parameters were as follows: The foaming temperature was 80~105 °C, the foaming pressure was 17 MPa and 23.8 MPa, and the time of CO_2_ impregnation was for 60 min. This impregnation time can make the CO_2_ in the material reach a saturated state.

### 2.5. The (SA/TPS)/PBAT Composite Foam Appearance and Internal Structure SEM Analysis

The foam samples after supercritical foaming were plated with gold on the cross-sectional surface by vacuum evaporation, and then the structure of the foam cells, the size of the bubbles, and the dispersion of TPS in the composite were observed by scanning electron microscopy (SEM, using a Hitachi TM4000 Plus, Hitachi High-Tech Fielding Corporation, Nagano, Japan), and the WD parameters were 15.6 mm, the HV voltage was at 5.0 kV.

### 2.6. Chemical Modification and Analysis of TPS Functional Group

At the interface of TPS and PBAT polymer substrate, adhesion is not good due to the viscous surface of TPS and the lipophilic polymer base. For improving the adhesion between TPS and PBAT polymer, surface treatment becomes essential. The addition of SA coupling agents provides the required adhesion property for their conjunction and lowers the interfacial shear strength of TPS to polymer substrates [47].

SA chain consists of the organic function and the alkoxy function at the terminals which interact with PBAT polymer and TPS to become more hydrophilic and improve the affinity with polymer [48].
(1)$$X-{\left(CH_{2}\right)}_{n}-Si{\left(OR^{\prime }\right)}_{3},\;n=0\sim 3$$

The ${OR}^{\prime }$ is a hydrolyzable alkoxy group, and the X is an epoxy-functional group. Silanes ${OR}^{\prime }$ end form a hydrogen bond with the surface of TPS containing hydroxyl groups. Moreover, the polymer treated with silane can improve wetting, and the organic functional group X in the coupling agent reacts with the polymer to form an interpenetrating network. Eventually, a formed network renders reinforcement between the TPS and the polymer interfaces. As shown in [49,50], the silyl group is first hydrolyzed and then condensed, forming a bond of silanol during acid-base condensation. In addition, the hydroxyl groups on the substrate surface of the TPS, and silanol form a polysiloxane structure. The melt strength of the TPS structure did not prevent the foaming gas from escaping into the environment and maintaining the structure of the pores. Therefore, using supercritical ${\mathrm{CO}}_{2}$ foam foaming technology in the development of starch foaming materials is facing challenges.

In this study, we synthesized the reactive functional groups at the ends of the thermoplastic starch molecular chain to produce a three-dimensional structure, so that the molecular chains can be entangled and intertwined with each other, and their viscoelastic and plasticizing behaviors achieve the strength requirements of the foam material structure. The process is shown in [49,50].

### 2.7. Supercritical Batch Foaming Process

Figure 1a schematically describes the processing system used for CO_2_ batch foaming in the study. An upper and under part of a batch die is mounted on the upper and under the platform of the press, respectively. A small tunnel in the batch die was connected with a CO_2_ injection unit by a stainless steel tube. The press can provide heat to raise the temperature of the batch die and force to seal foaming agent CO_2_ in the die. The inside dimensions for the batch die are 250 mm in diameter and 13 mm in height. The foaming procedure was as follows: (a.) PBAT with thermoplastic starch and silane were blended to form a large sheet of 3 mm in thickness. Next, the large sheet was cut into several small square samples of 25 mm in length and 3 mm in width thickness for foaming. (b.)The batch die was heated to reach setting temperatures ranging from 80 °C to 105 °C first; then, five square samples were placed in the under part of the die, and next, we sealed the upper and under part of the dying by the force from the piston of the press. (c.) Foaming agent CO_2_ was injected into the batch die by using a CO_2_ injection unit through a stainless steel tube until the setting pressure, which is 17 MPa or 23.8 MPa. (d.) The five square samples were soaked with CO_2_ under a supercritical condition for an hour to allow the samples to reach saturation conditions. (e.) The batch die was opened after an hour and the open rate of the die was instant less than 1 s by lowering the piston of the press. (f.) The five foam samples were taken out immediately from dying to prevent shrinkage from a high temperature. (g.) The foam samples were cooled at room temperature to let foam samples be stable. (h.) The properties and structures of five foam samples at the same foaming conditions were evaluated when foam samples were at room temperature after 24 h.

## 3. Results and Discussions

### 3.1. FT-IR Analysis of TPS and SA/TPS

Figure 1b shows the FT-IR spectra of the TPS polymer functional groups before and after the surface modification. The main IR peaks at 3381 cm^−1^, 2922 cm^−1^, 1651 cm^−1^, and 1401 cm^−1^ represent the O–H stretching, C–H and -CH_2_ asymmetric stretching, C=O stretching, and -CH_2_-deformation, while the peak position at 1030 cm^−1^ of starch may be caused by C–O–H stretching. The modified SA/TPS IR peaks are slightly shifted to 3397 cm^−1^, 2924 cm^−1^, 1652 cm^−1^, 1401 cm^−1,^ and 1033 cm^−1^, respectively, which represented the SA interaction with the TPS functional groups. Especially, the peak position is shifted from 1405 cm^−1^ to 1115 cm^−1^. The position of the corresponding peak of the SA modifier was with relatively low frequency. The SA modifier and this frequency are close to the corresponding starch where C–H is the -CH_2_- asymmetric expansion and C–O–C expansion and contraction.

#### H NMR Analysis of TPS and SA/TPS

The TPS NMR analysis spectrum is shown in Figure 1c. The sample is dissolved in DMSO-d_6_. Compared with literature data [48], there are small signals at 3.6 (H-6a, H-6b, and H-5), 4.6 (H-2), 5.08 (H-4), and 5.41–5.52 (H-1 and H-3) ppm. This indicates the resonance of starch. The small signals at 2.49 and 3.38 ppm are from the -OH (hydrogen bond and water) in DMSO and starch, respectively, and the resonances at 4.41 and 4.48 ppm are from the terminal proton -CH_2_-OH group, respectively.

The (SA/TPS) NMR analysis spectrum is shown in Figure 1d, where (PEG/TPS) is dissolved in DMSO-d_6_. This small signal is at 3.6 (H-6a, H-6b, and H-5), 4.6 (H-2), 5.08 (H-4), and 5.41–5.52 (H-1 and H-3) ppm, representing starch resonance [48]. The peaks at 2.49 and 3.38 ppm come from DMSO and the -OH group, and -OH (hydrogen bond and water) in starch. The resonances at 4.41 and 4.48 ppm come from the proton at the terminal functional group of -CH_2_-OH, and the 3.49 ppm signal comes from the resonance of the SA modifier proton at the -Si-OCH_3_ functional group.

### 3.2. The Composite Foam Density of TPS/PBAT

#### 3.2.1. The Density of Foamed Material of PBAT

Figure 2 shows the density changes of PBAT foams impregnated with CO_2_ at different foaming temperatures (80, 85, 90, 95, 100, and 105 °C) with two different foaming pressures (17 MPa and 23.8 MPa). The obtained results show that the foam density of PBAT was higher at a lower pressure than the high-pressure conditions. However, the pressure changes in PBAT foaming processes were not brought to any noticeable density differences. Consecutively, the temperature rises along with the above pressure condition and affects the PBAT foam density proportionally; furthermore, the obtained foam density was moved from 350 kg/m^3^ to 160 kg/m^3^.

#### 3.2.2. The Density of TPS/PBAT Composite Foam

Figure 3a represents the foam density of TPS/PBAT blends by impregnating CO_2_ at the foaming pressure of 17 Mpa and foaming temperatures between 80 °C and 105 °C. The three curves in Figure 3a indicate the trends of foam density for three blends, i.e., 70% TPS/30% PBAT [N-3], 60% TPS/40% PBAT [N-2], and 50% TPS/50% PBAT [N-1]. The form density of PBAT increases as the concentration of TPS increases in the TPS/PBAT composite. In particular, the forming density of [N-3] reaches its maximum at the forming temperature of 80 °C, which is approximately 1120 kg/m^3^. However, the increasing temperature was not ideal for the foam-forming process. Because of the high temperature, the microcellular structure of the composite form starts to shrink, which leads to poor elongation properties. It can be found that 50% of TSP shows a 20% elongation with the lowest foam density of 340 kg/m^3^. A low foam density of the three blends occurs around the foaming temperature of 100 °C, however, the foam density rises again when foaming temperatures are higher than 100 °C. TPS is a material that is not easy to foam, especially CO_2_ as a foaming gas, which is mainly used as a filler for biological materials.

Figure 3b shows the density change of TPS/PBAT composite foam obtained by impregnating CO_2_ with TPS/PBAT composite at a foaming temperature range of 80 °C to 105 °C and a foaming pressure of 23.8 MPa. In Figure 3b, curves are representing the foam densities of the [N-1], [N-2], and [N-3] composites. From the obtained results, the maximum foam density of the composites at 80 °C was reduced from 1120 to 700 kg/m^3^ because of the high-pressure environment; when the foaming temperature is 80 °C, compared with other foaming temperatures. Similarly, as the foaming temperature continues to rise, the density of all foams will continue to decrease until the foaming temperature is 95 °C. If the foaming temperature rises again, the density of the foams will also increase slowly. The bubble structure cannot be supported, causing shrinkage (as shown in Appendix A). It was found that the lower foaming density of this foaming pressure of 23.8 Mpa for the composite [N-1] is about 300 kg/m^3^, occurring at a foaming temperature of 95 °C, and the same foaming temperature varies as the TPS content increases. The foam density of the composite will also increase with the increasing TPS content. The TPS/PBAT composite system is more difficult to foam, however, the increase in foaming pressure will help the density of the composite foam to decrease. However, the bonding between TPS and PBAT was very poor, especially since the 80% TPS/20% PBAT blend is difficult to foam.

#### 3.2.3. The Density of (SA/TPS)/PBAT Composite Foam

Figure 3c shows the density change of the 50% (SA/TPS)/50% PBAT composite under the foaming temperature range of 80°C to 105°C, and the foaming pressure of 17 MPa impregnated with CO_2_. This experiment uses an SA modifier to modify the terminal functional groups of TPS, to verify whether it can promote the interface between TPS and PBAT to produce better compatibility. In Figure 3c, there were three curves representing the foam density of unmodified composite [N-1]: 50% TPS containing modifier SA addition and is 5 PHR (PHR: per hundred resin (TPS)) combined with 50% PBAT [S-0.5], and 50% TPS containing 10 PHR SA in TPS combined with 50% PBAT, [S-1]. Among them, [N-1] foam has the highest foam density, while [S-0.5] and [S-1] show a foaming temperature of 100 °C and have a lower foaming density of 180 kg/m^3^ and 160 kg/m^3^, respectively. It was proved that the SA concentration in TPS significantly modifies and improves physiological modification, such as tensile strength, elongation, and foam density. While the foaming temperature is higher than 100 °C, the density of the three kinds of composite foams all have an upward trend, however, the density of [N-1] foam is comparatively high, which also confirms the internal structure of the foam strength as poor.

Figure 3c,d used three types of the same composite and the same foaming temperature and impregnation gas, while Figure 3d used a higher foaming pressure of 23.8 MPa. The density of the three kinds of foam is under the same foaming temperature range, and the changing trend of the density of the composite foam is similar to Figure 3c. However, under higher foaming pressures, the density of the three types of composite foams is relatively low; the [N-1] compound and [S-0.5] compound have the same lower foaming. The low density of 300 kg/m^3^ and 200 kg/m^3^ occurred at the foaming temperature of 95 °C, whereas [S-1] shows the lowest foam density of 150 kg/m^3^ at a foaming temperature of 100 °C.

#### 3.2.4. The Influence of Foaming Temperature on the Density of Composite Foams with PBAT in Different Ratios (SA/TPS)

Figure 3 shows the foam density of SA (10 PHR) surface-modified TPS mixed with a PBAT polymer composite blend. At fixed pressure 17 MPa, the foam density of various mixing ratios of SA/TPS and PBAT was investigated under changing temperatures. Mixed weight ratios of 50%, 60%, 70%, and 80% SA/TPS were blended with 50%, 40%, 30% and 20% PBAT, as the mixed composites named [S-1], [S-2], [S-3] and [S-4], respectively. It can be found that the addition of 50% (SA/TPS) can obtain the lowest foam density of about 160 kg/m^3^ at a foaming temperature of 100 °C. When (SA/TPS) is added at 60%, 70%, and 80%, the lowest foam density appears at the foaming temperature of 105 °C, and the foam density is 250 kg/m^3^, 420 kg/m^3^, and 580 kg/m^3^, respectively. We found that increasing the concentration of the SA/TPS composite ratio in PBAT polymer was unfavorable to the foam composites, whereas the relative stability of the composites is also better, and therefore, increases the temperature of the lowest foaming density.

A similar procedure was followed in terms of the high pressure 23.8 MPa conditions, where it renders the four composites foam density changes values. Figure 3f shows that the ratio of (SA/TPS) in the composites is 50%, 60%, and 70%, and the lowest foam density is about 180 kg/m^3^, 220 kg/m^3,^ and 450 kg/m^3^. When the ratio of 80% (SA/TPS) is mixed, the lower foam density occurs when the foaming temperature is 105 °C, and the foam density is 520 kg/m^3^. It can be found from the research, however, that the (SA/TPS) ratio is 70% and 80% of the composites if the foaming temperature of this experiment is 100 °C. There is no obvious change in the foam density at a higher foaming temperature. When the foaming pressure is 17 MPa, the density of the two-component composites foam shows a downward trend rather than an upward trend. This means that the more the proportion of (SA/TPS) in the composites is, the higher the heat resistance of the composites and the smaller the change in foam density. However, the density of the foam is relatively high, which means that (SA/TPS) is similarly filled in the composites. It is of little help to the foaming process of the composites.

### 3.3. Appearance and Internal Structure of TPS/PBAT Composites Foam

#### 3.3.1. Appearance and Internal Microstructure of PBAT Foam

Figure 4 shows the change of the foam impregnated with CO_2_ at the foaming temperature of 95 °C and the foaming pressure (a) 17 MP and (b) 23.8 MPa.

For the appearance of the samples after foaming for 1 h, and stored at room temperature for 24 h, we were found that at the foaming temperature of 95 °C, the softer PBAT was found to be smooth and the foam surface was flat and elastic, however, the foam volume becomes larger as the foaming pressure increases.

After supercritical foaming, the foams were obtained from foaming pressures of 17 MPa and 23.8 MPa, and the initial appearance of the foam is without cracks. After 24 h of storage, the foam became deformed and smaller in volume, which caused the material damage. The reason, however, might be due to the carbon dioxide gas in the foam being released into the atmosphere faster than the air in the atmosphere penetrates into the foam, resulting in negative pressure created inside the foam during the cooling process; however, the foam structure cannot support this negative pressure, which causes the foam to shrink. Moreover, thermal expansion and contraction are also the reason. There was a small residual temperature inside the foam even after 2 h. Although, after more than 20 h of cooling, the temperature of the foam continues to decrease, causing the volume to continue to shrink.

Figure 4c is a SEM image of the internal microstructure of the CO_2_ foam impregnated with PBAT at a foaming temperature of 95 °C and a foaming pressure of 17 MPa. The depicted figures of PBAT foams were magnified by 50 times and 1000 times, respectively, and it was found that PBAT foam was achieved with uniformly distributed bubbles under this foaming condition. Most of the bubbles have a pentagonal or hexagonal cross-section and each bubble has obvious bubble walls, which means that the foam has a closed-cell structure. Since the internal bubbles of the foam have a complete structure, the surface is smooth, and the average bubble diameter is about 25 μm.

#### 3.3.2. The Appearance and Internal Structure of TPS/PBAT Composites Foam

The photographic images in Appendix A render the internal structure of the unmodified [N-1] composite after the CO_2_ foaming process at a temperature range of 80–105 °C and a fixed pressure of 17 MPa. We observed that the composite foams expand slightly, however, the surface properties remain smooth and hard due to their high foam density value at the foaming temperature of 80 °C. As the foaming temperature rises, the composite foam continues to expand. However, as the foaming temperature rises to 100 °C, the surface of the foam becomes uneven shapes that are relatively softer; therefore, raising the foaming temperature results in the lowest foam density. However, at the foaming temperature of 105 °C, the shape of the foam is completely distorted and rigid, the appearance of the foam surface becomes more rough and uneven, the flexibility of the foam deteriorates, and the volume is severely collapsed.

Appendix A shows the photographic images of the unmodified composites [N-2] and [N-3], respectively. It was found that at the low foam temperatures of 85 °C and 90 °C, the composite foams only slightly expand, whereas the surface properties remained unchanged. When the foaming temperature rises, the expansion of the [N-2] foam is slightly larger than that of the [N-3] foam. The appearance of the two foams is distorted and deformed while the foaming temperature rises to 100 °C and 105 °C. Among them, the [N-3] foam has no flexibility, and the surface of the foam is rough, which is close to no foaming. Whereas, performing the same process under 23.8 MPa pressure conditions, the [N-1] foam volume is comparatively larger than at 17 MPa pressure conditions. Similarly, the surface of the [N-1] foam becomes a more rough and uneven shape. The photographic image was included in the Appendix A. The other two composites, [N-2] and [N-3], also produced a similar trend, as seen in Appendix A. Although, at this higher foaming pressure, the softness of the foams was slightly better than the composites at the low foaming pressure of 17 MPa. However, the overall softness of the foam is still very poor, the surface is rough, and the density of the foam is still relatively high.

Figure 5a–c renders the cell structures of the [N-1], [N-2], and [N-3] foams made at the foaming temperature of 95 °C and the fixed foaming pressure of 17 MPa. Many large cells or pores and small TPS particles scattered in the foam were observed in Figure 5a. Large cells, more than 100 μm, and small cells of less than 20 μm coexist in the foams; furthermore, the cell wall is not intact and has tiny holes. Unreacted TPS particles can be seen clearly and the boundary between the TPS particles and cells is also very distinct. Moreover, the bubble size distribution of this foam is not uniform, where the diameter of the large bubbles was 10 times bigger than the small bubbles. Using the higher magnification SEM image, the bubble structure of the foam cannot show complete bubbles like the foam of PBAT in Figure 4c can. The bubble structure of this foam is inconsistent and the bubble wall is broken or merged with the neighboring bubbles, resulting in a large size pore volume. This is called the open-cell structure of the bubbles. Furthermore, the bubbles of the PBAT composites foam were affected by the undissolved TPS particles. Therefore, the degree of open-cell structure of the foam was much higher than that of the PBAT foam.

The cell structures for the [N-2] and [N-3] foams are shown in Figure 5b,c, respectively. It is not surprising that the addition of untreated TPS into PBAT improves the foam density. However, an increased concentration of TPS results in the undissolved particle agglomeration in the PBAT composite, such as the [N-2] and [N-3] composites showing 50 μm and 10 μm large and small pore volumes, respectively. Meanwhile, there are fewer incomplete bubbles in the foam were observed. Therefore, the obtained SEM image shows the uneven surface with a high open-cell structure, which causes the hardness and rigidity of the foams. It was figured that in the TPS/PBAT foam-forming process under supercritical CO_2_, the formed foam cells were particularly dependent on the PBAT rather than TPS. In Figure 5, it is proved that in the [N-1], [N-2], and [N-3] composites formed by unmodified TPS with PBAT, TPS seems to be used as a filler and has poor compatibility with PBAT. The mechanical properties of the composite will also deteriorate.

### 3.4. Appearance and Internal Structure of (SA/TPS)/PBAT Composites Foam

#### 3.4.1. The (SA/TPS)/PBAT Composite Foam Appearance and Internal Structure

To improve the interfacial diffusion between the TSP and PBAT polymers, a surface grafting regent (SA) was introduced. Initially, two various fractions of SA were used to modify the surface of 50% TPS, such as [S-0.5] and [S-1]. Appendix A’s photographic images of [S-0.5], and [S-1] are shown. The composite foams were produced under the same operating temperature and CO_2_ foaming gas was passed at a fixed pressure of 17 MPa. The foam sample become softer and improved the elastic behavior. Among the series of foams, foaming temperatures of 95 °C and 100 °C formed samples exhibiting a large pore volume with the lowest foam density. However, we have noticed a slight surface roughness on these foams. When the foaming temperature continuously rises to 105 °C, the structures’ appearance of foams for (a) [S-0.5] and (b) [S-1] are slightly deformed. However, when compared to the unmodified TPS/PBAT foams, this SA-modified TPS/PBAT provides better tensile strength and elongation properties.

At the same operational temperature, the foaming process was elevated to 23.8 MPa. The obtained results reveal the uneven surface as per the temperature raise (Appendix A). Furthermore, the large volume of foams occurred between a 90 °C to 100 °C temperature range; however, this resulted in foams becoming soft with slight surface roughness. When the foaming temperature is 105 °C, the volume of the foam tends to shrink. Thus, the foaming pressure seems to tend to increase the volume of the composites foam at the same foaming temperature, however, it is not conducive to high-temperature foaming at 105 °C. Appendix A shows the cell structure of [S-0.5] and [S-1] composite foams at fixed parameters of 95 °C and 17 MPa foaming temperature and pressure. The surface of the TPS particles in the [S-0.5] foam (as shown in Appendix A) is rougher than the surface of the TPS particles in the [N-1] foam (as shown in Figure 5), such as the surface of the unmodified TPS mixed foam. Although the TPS particles were modified with 5 PHR SA, the interface between this particle and PBAT was not clear. The TPS particles are uniformly dispersed inside the foam, however, the particle size is not uniform. The diameter of the TPS particles presented is less than 25 μm or greater than about 200 μm. Although the 5 PHR SA can increase the compatibility of TPS and PBAT, it limits the dispersion of TPS in PBAT. Most of the bubbles have a closed-cell structure; therefore, the bubble wall between the bubbles is obvious, and consecutively the bubble size distribution is relatively consistent. The SEM image shows that the average bubble diameter is about 10 μm, which is also the performance of good compatibility between TPS and PBAT.

The cell structure in Appendix A shows that the boundary between TPS and PBAT becomes fuzzier when compared with Appendix A; especially with the distinct small (TPS with SA) particles from PBAT being difficult, and the surface of the (TPS with SA) particles is surrounded by many small cells that are less than 10 μm in diameter. Most of the cells in Appendix A’s foam are closed cells. The phenomenon can be explained by the compatibility and miscibility in [S-1] having dramatically enhanced. However, large pores are also observed in Appendix A, which indicates that compatibility decreases between large (TPS with SA) particles and PBAT. Therefore, the dispersion of (TPS with SA) in PBAT is also an important factor.

#### 3.4.2. The Appearance and Internal Structure of (SA/TPS)/PBAT Foams with Different (SA/TPS) Ratios

Figure 6 shows the appearance of foams with different ratios (10 PHR SA/TPS) and PBAT composite at three foaming temperatures (at 95 °C, 100 °C, and 105 °C) and a foaming pressure of 23.8 MPa impregnated with CO_2_. After the surface-modified 10 PHR SA/TPS mixture was mixed with different ratios of PBAT to form the composite foams, such as [S-2], [S-3], and [S-4], it was observed that the volume of the foam decreased as the proportion of 10 PHR SA/TPS increased. This study was carried out at three foaming temperatures. As the foaming temperature rises, the volume of the foam increases. The volume expansion changes of [S-2] were the most obvious. However, for [S-4], the volume of the foam only slightly increased. This phenomenon proves that, when the TPS content reaches 80%, the interface between TPS and PBAT is still intact from the internal structure of the foam. Therefore, it is further confirmed that the addition of a modifier provides more compatibility between TPS and PBAT. When the proportion of TPS in the composite material is higher, it is easier to disperse in the composite material. Because the more reactive functional groups, PBAT and TPS are easier to connect with each other, the reaction makes the compatibility better.

The SEM image shows that the TPS in the composite foams becomes a continuous phase, the large pores become less, and more TPS particles with extended shapes are found, which is proof of the better compatibility between the two. However, the unmodified TPS (such as N-3) composite material with 70% TPS content exhibits poor mechanical properties, whereas the SA-modified TPS foam has excellent flexibility after being folded even at higher concentrations. The very different material properties before and after foaming are illustrated.

Moreover, Appendix A shows the appearance of the foam under the foaming temperature of 105 °C and different foaming pressures of 17 and 23.8 MPa impregnated with CO_2_, as shown in Appendix A, for [S-2], [S-3], and [S-4], respectively. At the same foaming temperature under 23.8 MPa, the three types of composites have a larger foam volume than the foaming pressure of 17 MPa. However, the surface of the foam deteriorates and large bubbles were generated. Whereas, the high proportion [S-4] foam provides a smaller volume than flat surface foam; although [S-4] has the least amount of foaming volumes, and the foam has flexural limitations, as shown in Figure 6d. The un-foamed composite board is easy to break, but the foamed board is not easy to break and has advantageous flexibility after bending, which also illustrates the different material properties after foaming.

The surface morphology change of the [S-1], [S-2], [S-3], and [S-4] composites at a low foaming pressure of 17 MPa with a fixed foaming temperature of 95 °C were depicted in the SEM image (Figure 7). Figure 7a shows that the composites [S-1] and [S-2] foams have similar internal structures with large pores, but the [S-1] foam contains smaller bubbles. Although these two foam shows a closed-cell structure without particle agglomeration, the bubble diameter of the [S-2] foam is about 15~20 μm. In addition, compared with the unmodified [N-2] foam (as shown in Figure 5b), the existence of complete bubbles cannot be seen.

Figure 7b shows that [S-3] has large pores inside the foam and tends to decrease. Besides the large holes, the bubble size is still uniform. However, the shape of the bubble is slightly deformed, and it is no longer the pentagonal or hexagonal shape of normal foam. The bubbles are mostly closed-cell structures, and the interface with PBAT remains intact without gaps, which can be judged as good compatibility. Because of the high fraction of surface-modified SA/TPS, the foam expanded with more bubbles, and the foam bubble diameter is about 15–20 μm, even though the un-foamed areas were also found.

When combining the SA/TPS with PBAT, the functional group of SA forms a cross-link with PBAT and reduces the pore size, whereas the unmodified TPS with PBAT foam exhibits large pores with a broken cell structure.

Many (SA/TPS) large particles are not spherical (instead, it is a special shape, as shown in Figure 7b). Because of the compatibility of TPS with PBAT after modification; it appears to be one phase by SEM. However, surface unmodified TPS is incompatible with PBAT and obtained SEM images appeared with two phases. Many (SA/TPS) large particles are not spherical and are typically incompatible.

Figure 7c shows that the surface of [S-4] foaming still occurs in the composite; here, the size of the bubbles inside the foam is more delicate, and compared with the other foams, it has fewer larger pores. However, the shape of the bubbles is deformed and the bubbles are mostly closed-cell structures, and the bubble diameter is about 10~15 μm. The [S-4] composites solution shows a complete dispersity of SA/TPS due to the high SA content. As the (SA/TPS) material has more reactive functional groups, it is easier for PBAT and (SA/TPS) to contact each other to cause a reaction and better compatibility. Figure 7c shows more (SA/TPS) particles of special shape and fewer large pores in the foam, which is proof that PBAT and (SA/TPS) have better compatibility.

### 3.5. Mechanical Properties of the TPS/PBAT and the (SA/TPS)/PBAT Composites Foam

Table 2 show the ultimate tensile strength and elongation break and foam density for the unmodified foams [N-1], [N-2], and [N-3] and the surface-modified foams [S-1], [S-2], [S-3], and [S-4], respectively. These various composite foams are produced at the foaming temperature of 95 °C and the foaming pressure of 17 MPa. The unmodified TPS/PBAT composites have approximately 4% of the ultimate tensile strength and elongation at the break for these three TPS/PBAT foams and become much shorter. However, the surface-modified foams show increased ultimate tensile strength, such as the [S-1], [S-2], and [S-3] blended foams as being about 4 times, 2.5 times, and 2 times larger than that of the [N-1], [N-2], and [N-3] blended foams, respectively, although, the elongation at break becomes decreased. The foam density of the (TPS with SA)/PBAT blend has the same trend as that of the TPS/PBAT blend. In general, the ultimate tensile strength of the (TPS with SA)/PBAT blends is basically better than that of the TPS/PBAT blends; however, the two types of blends generally have a similar elongation at break (shown in Table 3).

## 4. Conclusions

This research has successfully developed a highly flexible and compatible starch-based foam using supercritical carbon dioxide. This research completely covers the foam-forming optimization steps, including various foaming temperatures under two different pressures. Furthermore, blending ratios of the surface-modified (SA/TPS) and unmodified (TPS) starch were optimized with the PBAT foam. The results show that with the addition of surface-modified SA/TPS starch to PBAT, the composite foam turns into a soft altered foam density with improved elongation strength and tensile properties for flexible application. We also observed that the unmodified (TPS/PBAT) composite is not suitable for the foam to form due to its poor interface formation with PBAT. However, when compared to the internal microstructure, the SA/TPS/PBAT composite foam shows a uniform bubble with an average diameter of 25 μm and a closed-cell structure. Therefore, (TPS with SA)/PBAT blended foams exhibit lower foam density and better tensile properties than those of TPS/PBAT blended foams. In addition, this research can be applied to thermoplastic starch biodegradable foaming materials, combined with supercritical foaming green process technology, to develop starch composite foaming materials. This foam-forming processing technique could be used in electronics packaging materials and has medical equipment application potential in the future.

## Figures and Tables

**Figure 1 polymers-14-01952-f001:**
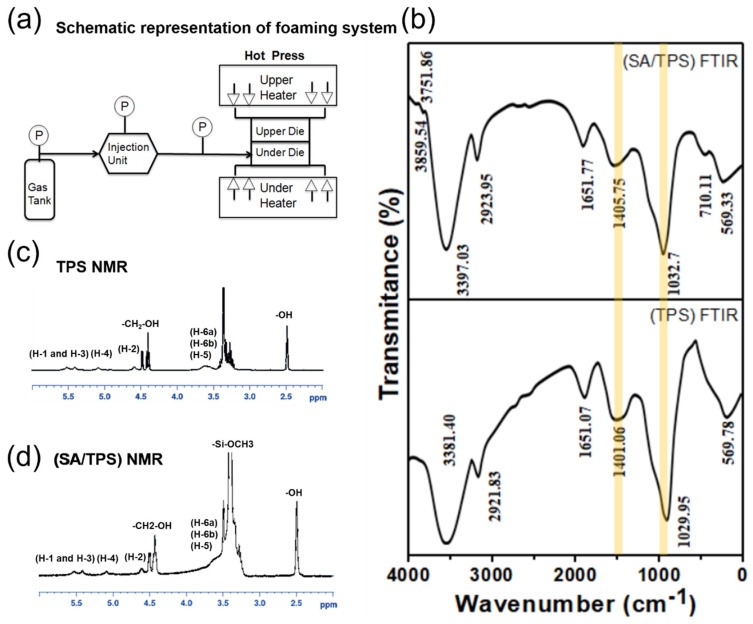
Schematic representation of foaming system and chemical modification for analysis of TPS functional group. (**a**) The processing system is used for CO_2_ batch foaming. (**b**) FT-IR of TPS and (SA/TPS). (**c**,**d**) NMR of TPS and (SA/TPS).

**Figure 2 polymers-14-01952-f002:**
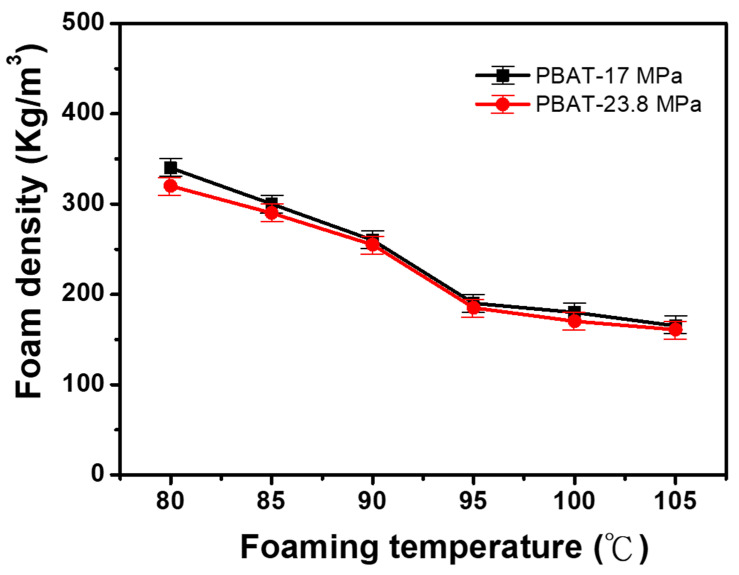
Dependence of PBAT foam density on six foaming temperatures and two foaming pressures.

**Figure 3 polymers-14-01952-f003:**
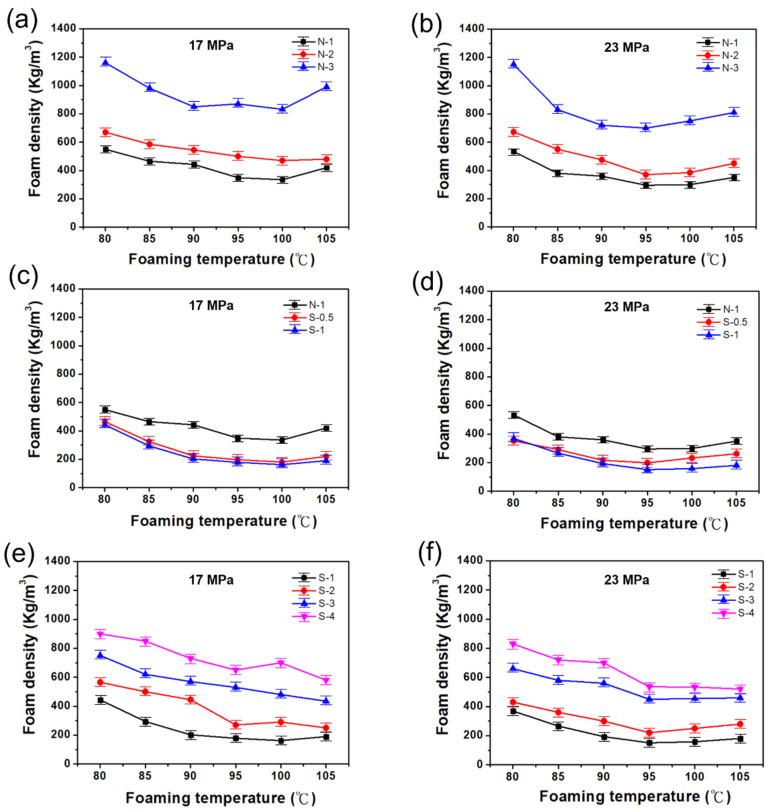
Dependence of TPS/PBAT and (TPS with SA)/PBAT composite foam density under six foaming temperatures and different foaming pressures. (**a**,**c**,**e**) Variation of composite foam density at 17 MPa. (**b**,**d**,**f**) Variation of composite foam density at 23 MPa.

**Figure 4 polymers-14-01952-f004:**
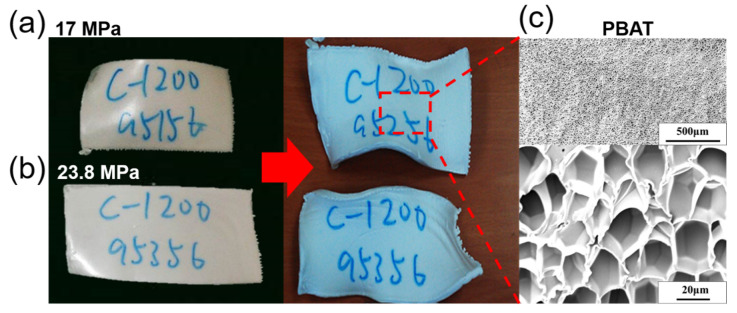
The changes in the foam impregnated with CO_2_ at the foaming temperature of 95 °C and the foaming pressure (**a**) 17 MP and (**b**) 23.8 MPa. (**c**) The SEM image of the PBAT foams was magnified by 50 times and 1000 times, respectively, at a foaming temperature of 95 °C and a foaming pressure of 17 MPa.

**Figure 5 polymers-14-01952-f005:**
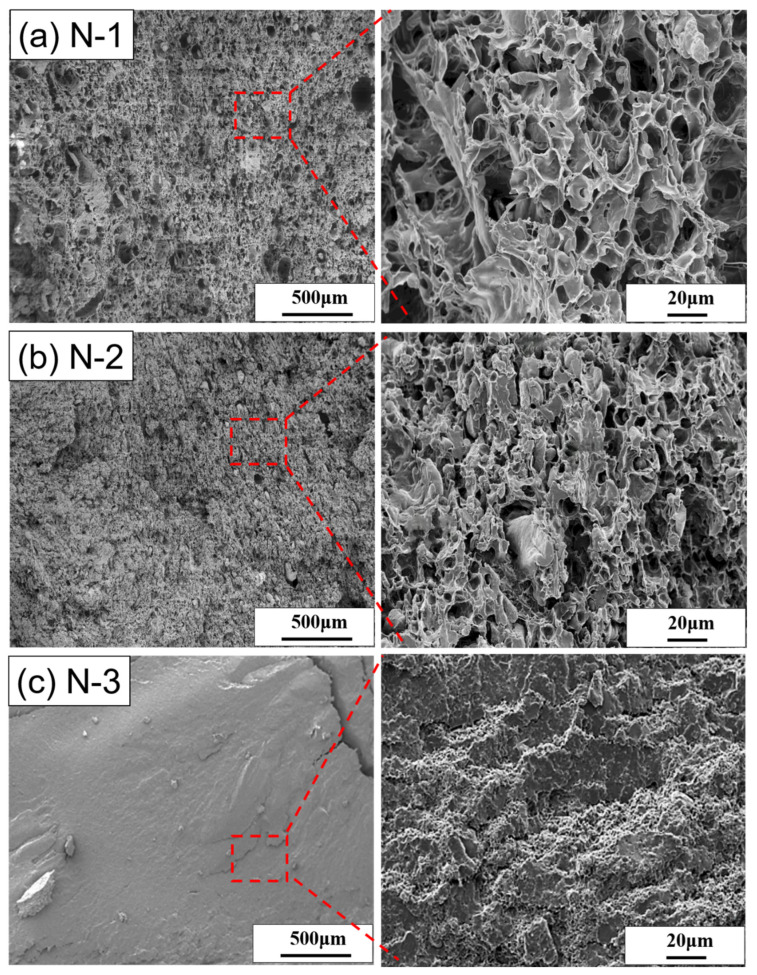
The SEM images of cell structure (**a**) [N-1], (**b**) [N-2], and (**c**) [N-3] at the foaming temperature of 95 °C and the foaming pressure of 17 MPa.

**Figure 6 polymers-14-01952-f006:**
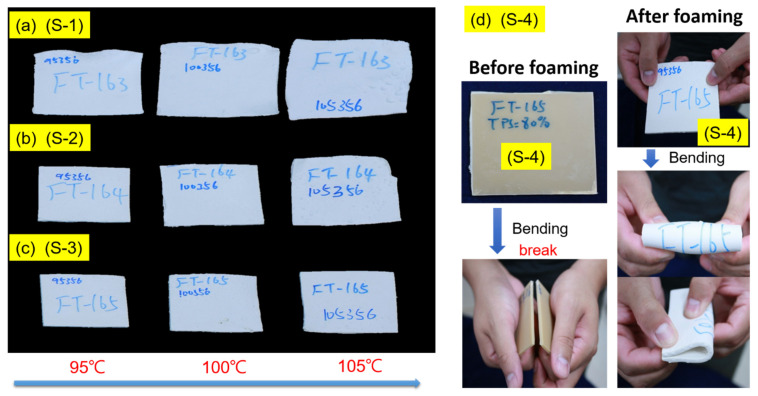
The appearance of foams with different ratios (SA/TPS) and PBAT composite (**a**) [S-1], (**b**) [S-2], and (**c**) [S-3] at three foaming temperatures and a foaming pressure of 23.8 MPa impregnated with CO_2_. (**d**) Comparing the foam appearance characteristics and flexural limitations of the [S-4] before and after.

**Figure 7 polymers-14-01952-f007:**
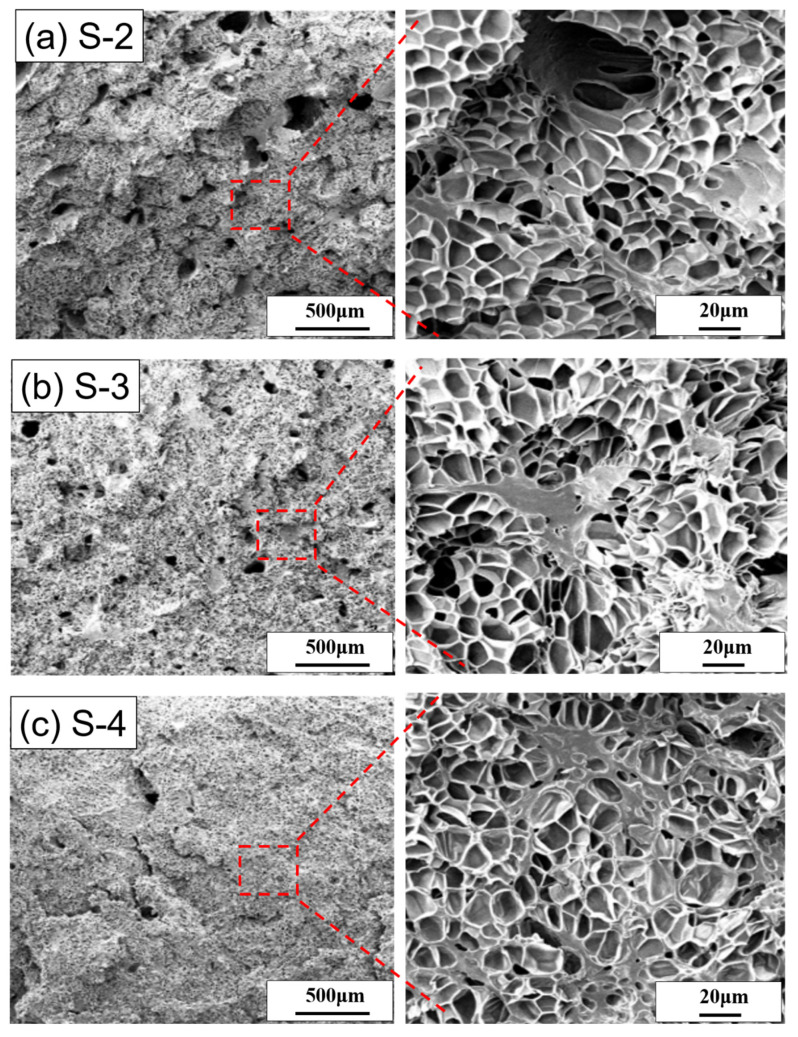
The SEM images of cell structure at the foaming temperature of 95 °C and the foaming pressure of 17 MPa. (**a**) [S-2], (**b**) [S-3], and (**c**) [S-4].

**Table 1 polymers-14-01952-t001:** The composite material composition ratio and symbols corresponding list.

Symbols	Blends (by Weight)
[N-1]	50% TPS/50% PBAT
[N-2]	60% TPS/40% PBAT
[N-3]	70% TPS/30% PBAT
[S-0.5]	50% (TPS with 5PHR SA)/50% PBAT
[S-1]	50% (TPS with 10PHR SA)/50% PBAT
[S-2]	60% (TPS with 10PHR SA)/40% PBAT
[S-3]	70% (TPS with 10PHR SA)/30% PBAT
[S-4]	80% (TPS with 10PHR SA)/20% PBAT

**Table 2 polymers-14-01952-t002:** The mechanical properties of the (SA/TPS)/PBAT composites foam.

	[N-1] (F)	[N-2] (F)	[N-3] (F)	[S-1] (F)	[S-2] (F)	[S-3] (F)	[S-4] (F)
Tensile strength (kPa)	226 ±35	392 ±44	539 ±48	883 ±45	1030±52	1157±40	1236±55
Elongation (%)	20±3	9±1	2±1	37±3	22 ± 2	13±2	10±1
Foam density (kg/m^3^)	349±2	540±1	897±1	187±3	287±2	515±2	663±1

**Table 3 polymers-14-01952-t003:** The mechanical properties of the TPS/PBAT composites.

	[N-1]	[N-2]	[N-3]	[S-1]	[S-2]	[S-3]	[S-4]
Tensile strength (Breaking point) (kPa)	226±35	392±44	539±48	883±45	1030±52	1157±40	1236±55
Elongation (%)	20±3	9±1	2±1	37±3	22±2	13±2	10±1

## Data Availability

Not applicable.

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
