# Peer review of "Thermoplastic Starch with Poly(butylene adipate-co-terephthalate) Blends Foamed by Supercritical Carbon Dioxide"

_polymers, 2022, doi:10.3390/polym14101952_

Round 1
Reviewer 1 Report
This work has studied various experimental parameters that affect the foams properties, such as the polymer solution concentration, foaming pressure, foaming temperature, and the characteristics of TPS with compatibil-izer /PBAT blends, as well as the foaming process conducted by supercritical CO2. However, the current form of this work is more like a research report rather than a scientific research paper. The nolvety of this paper also do not reach the standar of polymers. More detailed discussion are needed and the paper must be significantly improved before it can be acceptable.
- English and the style of scientific presentation should be improved. e.g. 50%, 40%, 30, and 20% biodegradable polyester (PBAT) should be 50, 40, 30, and 20 %.
- many discription of the results, such as "Figure 8 shows the appearance of foams with different ratios (10 PHR SA/TPS) and PBAT composite at three foaming temperatures (at 95°C, 100°C, and 105°C) and a foaming pressure of 23.8MPa impregnated with CO2. After surface modified 10 PHR SA/TPS mix-ture was mixed with different ratio of PBAT to form the composite foams, such as [S-2], [S-3], and [S-4]. It was served that the volume of the foam decreases, as the proportion of 10 PHR SA/TPS increases. This study was carried by three foaming temperatures. As the foaming temperature rises, the volume of the foam increases. The volume expansion changes of [S-2] were the most obvious. However, [S-4] the volume of the foam only slightly increases." there were discussions, but readers may interests to see what's the reasons to obtain the difference results.
- as starch-based biodegradable foams, are the foams prepared by this study biodegradable? Where can the foams be potentially used and how is their performances?
Reviewer 2 Report
Dear Authors,
Overall, this is an interesting manuscript that probably required a lot of work. In addition, TPS starch materials are interesting materials that still require improvement. However, in order for this manuscript to be published, it is necessary to introduce some important corrections.
Detailed comments below:
Beginning of the introduction, paragraph 4: I think it is worth mentioning here the TPS pouring method for Teflon molds. Although these are mainly research methods, they are widely used. In addition, you should write about the use of various (other) biodegradable additives used to strengthen TPS. This opens up great opportunities for the production of innovative biodegradable products for really different applications in the agro-food industry. Don't limit yourself to packaging only. TPS-based materials can have many applications. Please see the article below: "Properties of Biocomposites Produced with Thermoplastic Starch and Digestate: Physicochemical and Mechanical Characteristics". Of course, more typical additives are also used: "Bionanocomposite films developed from corn starch and natural and modified nano-clays with or without added blueberry extract". And other.
At the end of the introduction, write down what your research will contribute to the development of biodegradable materials. Also, emphasize the innovation of your research more. The purpose of the work should also be improved. Try to write it more scientifically.
Methodology: all raw materials used in the research should be described as follows, (name of the raw material: producer, city, country). Describe the research equipment in a similar way.
Add more scanning microscopy parameters.
Write down how many repetitions of the experiments performed were made.
Chapter 3.1 and 3.1.1. These chapters are written just like the rest of the research methodology. These are, however, descriptions of research results. You should correct (edit) this part of the work and make a correct description of the results obtained.
Or maybe it should be a further part of the methodology? Consider it.
Figure 2 and 3. In the graphs you should add error whiskers. There may also be a standard deviation.
Table 2. In this table, the standard deviation should also be added for each result (+/- …….).
In my opinion, the conclusions are written quite well. However, review the results point by point again and see if they cover all the results.
Reviewer 3 Report
Starch-based biodegradable foams with high starch content were developed using industrial starch as base material and supercritical CO2 as blowing or foaming agents. Superior cushioning properties of these foams can lead to competitiveness in the market. The paper is interesting and could be accepted after the revision.
-The authors used a mixing machine for preparation of mixtures. Why an extruder was not used in order to get more homogeneous mixtures ?
-Range of y axis of Figure 2 should be from 0 to 500.
-It seems that Figures 5 and 7 do not provide a useful scientific information and could be removed ?
-It is stated that the described technology could be used in electronics packaging materials and in medical equipment application. Could properties of the described materials compared with industrial materials used in the mentioned fields ?
Round 2
Reviewer 1 Report
The authors have revised the manuscript by considering the suggestions and it has been improved. But they just corrected the places I have pointed out. Other similar flaws are not correct. Such as:
(1) The main IR peaks at 3381 cm-1, 2922 cm-1, 1651 cm-1, and 1401 cm-1 in section 3.1;
(2)The superscript and subscript of the units must be double checked.
Reviewer 2 Report
Dear Authors,
Thank you for adapting to the comments in the review. I accept all corrections.
A little note:
Note the word (Basel) in a few references - it's not needed. You should remove this word. e.g. reference 7, 14, 15, 18, 27, 29, 35
